# OpenReview forum: "Digital Organ Twins from Single Imaging Scans: A Hybrid Physics-ML Framework for Predictive Medicine"
_EurIPS.cc/2025/Workshop/MedEurIPS — EurIPS 2025 Workshop MedEurIPS Submission_

### Official Review · Reviewer_nKvc · 2025-10-24
**Review for Digital Organ Twins from Single Imaging Scans: A Hybrid Physics-ML Framework for Predictive Medicine**

**Rating:** 3
**Confidence:** 4

**Review:**

Summary:
This paper introduces OrganTwins, a framework that aims to construct digital organ twins from single medical scans through a hybrid physics–machine learning approach. It combines neural ODEs with physics-informed constraints to model organ dynamics. However, the manuscript omits many key implementation details, making it difficult to understand how the method actually functions in practice.

Strengths:
The integration of physical modeling and digital twin concepts is interesting and timely, representing a potentially valuable direction for computational medicine.

Weaknesses:
The paper lacks clarity and critical methodological information. Experimental tables appear inconsistent with the textual descriptions, and Figure 1 is confusing—its subfigures are crowded and poorly labeled, leaving the reader uncertain about what is being illustrated.

---

### Official Review · Reviewer_TcHo · 2025-10-24
**An interesting approach, but the method could be explained better.**

**Rating:** 6
**Confidence:** 4

**Review:**

Strengths
-	Interesting novel approach.
Weaknesses:
-	Page limit is exceeded.
-	The explanation of the method is not very precise (2.3). I admit the space constraints, but it would be possible to reference similar works. i.e. which neural ODE architecture was used?
-	Training procedure is also not explained. Are multiple time points needed for training?
-	Baselines are not discussed at all. Not even references are provided.
-	In Fig. 1b the image does not change.
-	Code is not yet released.
-	Figure way to small

---

### Decision · Program_Chairs · 2025-10-31

**Decision:**

Reject

**Comment:**

Both reviewers find the idea of creating digital organ twins from single scans original and timely, but they note significant gaps in methodological clarity and missing implementation details.